# Teachers’ Language Use in Multilingual Head Start Classrooms: Implications for Dual Language Learners

**DOI:** 10.3390/children9121871

**Published:** 2022-11-30

**Authors:** Megan Chan, Maria Belen Buttiler, Francis Yang, Jerry Yang, Yuuko Uchikoshi, Qing Zhou

**Affiliations:** 1Department of Psychology, University of California, Berkeley, CA 94720, USA; 2School of Education, University of California, Davis, CA 95616, USA; 3Kai Ming Head Start, San Francisco, CA 94103, USA

**Keywords:** DLLs, preschool teachers, language use, Head Start, multilingualism

## Abstract

Dual language learners (DLLs) are sensitive to teachers’ language influence in early childhood classrooms. In this mixed methods study incorporating 53 teachers from 28 preschools in Northern California, we investigated the characteristics of teachers’ language use in preschools teaching Chinese–English and Spanish–English DLLs. We further examined the links of teachers’ language use to the DLLs’ expressive vocabulary in English and their heritage language (HL), controlling for home language exposure and other confounding variables. Finally, we conducted interviews with teachers to understand how they make meaning of their daily language practices. The sample of children consisted of 190 Chinese–English (*N* = 125) and Spanish–English (*N* = 65) DLLs (mean age = 48.3 months; 48% females). The teacher survey showed that most teachers spoke two or more languages and used a mix of English and their HL during their interactions with DLLs. The results of random-intercept models showed that teachers’ language use did not uniquely predict children’s vocabulary, controlling for family-level factors. However, the teachers with more years of teaching DLLs oversaw children with a higher HL vocabulary. The interview data revealed that teachers employ several strategies to communicate with DLLs and support HL maintenance. Our study reveals the multilingual backgrounds of preschool teachers and the rich language strategies they implement in multilingual classrooms. Future directions concerning the quality and development of teachers’ language use are discussed.

## 1. Introduction

Teachers can help young children develop language and literacy skills during the preschool period. Dual language learners (DLLs), or young children under the age of 8 who are learning two or more languages at home and school [1], are particularly sensitive to language influences by teachers [2,3]. Indeed, recent research has reported positive associations between the quality of teachers’ talk and the language outcomes of dual language-learning children [4]. Young DLLs constitute one of the fastest-growing populations, making up nearly one-third of all young children between the ages of 0 and 8 in the U.S. [1]. California has the highest number of DLLs compared to any other state, as nearly 60% of Californian children are bilingual [5]. During preschool age, many DLLs, especially those from low-income families, attend federally and state-funded Head Start centers. These centers offer early childhood education programs to promote school readiness for children from birth to 5 years of age. The Head Start programs deliver services to low-income children and their families in core areas of early learning, health, and family well-being [6]. Consistent with the research suggesting the benefits of heritage language (HL) development for promoting DLLs’ cognitive, academic, and socioemotional readiness for school [7], current Head Start policies support the use of DLLs’ HL in early childhood classrooms [8]. Given the diverse linguistic practices of the families and teachers, DLLs may be exposed to varying levels of English as well as other languages. For example, DLLs in the U.S. start learning English as a second language while they are still acquiring their first language, also known as a home or heritage language [1]. It is important to examine the diverse linguistic backgrounds and language practices of the teachers that DLLs are exposed to in early childhood classrooms. Past research on teachers and early childhood settings has focused on monolingual English-speaking [9] and Spanish-speaking preschoolers [10,11,12,13]. One study compared preschoolers’ language outcomes in transitional Spanish–English bilingual programs (where teachers initially used a higher proportion of Spanish for instruction, gradually transitioned to a mix of Spanish and English, and eventually transitioned to a higher proportion of English) with those in monolingual English-speaking Head Start programs. Compared to the children in monolingual programs, preschoolers in transitional bilingual programs had an increased receptive and expressive vocabulary in Spanish without a cost to their English vocabulary [10].

To the best of our knowledge, there have been no studies examining teachers’ language instruction in Head Start classrooms serving Chinese-speaking DLLs. One study on complementary schools (i.e., voluntary schools for specific cultural groups) in the UK found that bilingualism and code switching among Chinese-speaking teachers and students promoted multilingual creativity, thus benefiting bilingual Chinese immigrant students academically [14]. Another study examined Chinese parents’ emotion talk during book-reading time and found that parents consistently used basic emotional language such as happy, sad, and angry in Cantonese [15]. Although the home language emotional environment can differ from that of classroom settings, the findings from Huang and Kan have implications for how Chinese bilingual teachers can utilize Cantonese instruction in the classroom (e.g., during book reading).

The current study addresses these gaps in the literature by examining preschool teachers’ demographics, the teachers’ classroom language use, and their links to DLLs’ expressive vocabulary in Head Start classrooms teaching DLLs from mainly Chinese-speaking and Spanish-speaking homes.

### 1.1. Theoretical Background

As originally explained by Bronfenbrenner (1977) in the bioecological systems model, there is a microsystem that entails the complex relations between the child and their environment [16]. This microsystem includes not only the child’s family, but other contexts such as preschool [17,18]. In these contexts, teachers and students interact with each other through various communication practices, which may entail the use of different languages. Empirical evidence has suggested that teachers, who are part of the school microsystem, have a critical influence on DLLs’ language development [19]. A recent meta-analysis of teachers’ language practices in early childhood classrooms [20] showed that teachers who used more linguistically complex speech tended to use an increased amount of teacher talk. Additionally, teachers who were more responsive to child talk asked more questions and engaged in more interactive conversations with children than the teachers who were less responsive to child talk. Importantly, teachers’ language practices have been positively associated with DLLs’ language and literacy development. A study conducted by Reilly et al. (2020) observed that teacher-directed activities were related to higher literacy skills among DLLs in Head Start classrooms [21].
*The integrative risk and resilience model* [22] *builds on the bioecological systems model and helps to better understand the immigrant DLL-specific context. It highlights the resources available, the quality of the curriculum, the pedagogical practices, and the supportive relationships with peers and teachers that can promote DLLs’ academic motivation and learning. Based on these frameworks, how preschool teachers in multilingual classrooms interact with DLLs may be associated with better developmental outcomes among DLLs.*

### 1.2. Teachers’ Language Exposure to DLLs

A significant portion of the DLL population in the U.S. comes from low socioeconomic status (SES) households and backgrounds [23,24,25]. DLLs from low-SES families are more likely to have a parent with limited or no English proficiency [25] and may have more restricted access to books and other literacy resources when compared to children from higher-SES families [26]. Thus, these children depend more on school programs to enrich their early language and literacy experience [27]. Consequently, teachers play a crucial role in creating equitable language environments in which to help DLLs develop socio-emotional and academic competencies [28,29,30]. Hence, it is critical to gain a better understanding of the demographic, language abilities, and classroom language use profiles of preschool teachers in Head Start programs serving the DLL population.

Previous research has revealed variability in the literacy environment of preschool classrooms, which is associated with teachers’ language use and the preschool curriculum [27,31]. For example, studies have shown that teachers vary with respect to how they teach children words and sounds, use language with children [32], conduct literacy activities [31], read books with young children [33,34], and integrate writing into the classroom [35]. Dickinson and colleagues (2014) showed that differences in teachers’ language use in the classroom varied by both the setting and type of activity that they were engaging in [33]. For example, academic English language use was prominent during book-reading time and group content instruction. However, this study was conducted in Head Start centers serving mostly African-American children in which English was the only language of instruction.

Moreover, teachers’ choice of language use in the classroom has been shown to be associated with DLLs’ school readiness. Miller (2017) collected classroom observations in Head Start classrooms and found that Spanish was always used to help promote English language skills, academic skills, executive function skills, and socio-emotional development [24]. For example, large group literacy activities, such as sing-alongs and story time, were always exclusively in English, followed by a direct Spanish translation. Teachers used both English and Spanish interchangeably when asking questions regarding executive function-related skills of planning and recall (e.g., “what do you plan to do during center time?”). Teachers tended to use more casual sentences in both English and Spanish when providing emotional support to children. Collins (2014) found that the type of instruction program played a role in children’s language development [36]. Specifically, Latino children who received instruction in both English and Spanish achieved higher proficiency levels in both languages compared to those who only received instruction in English. However, Baker and Páez (2018) found similarities in teaching practices among teachers and children who share a home language and teachers and children who do not share a home language [37]. In this study, most of the students in Head Start programs were DLLs with diverse home languages including Albanian, Amharic, Arabic, Haitian Creole, Portuguese, Spanish, and Somali, and all the teachers shared the idea that bilingualism should be considered an asset.

In terms of methodology, the majority of past studies used quantitative approaches—mostly correlational analyses—to explore language and literacy practices in preschool settings [27,31]. In contrast, Clark and Kragler (2005) used a mixed methods approach to investigate the effects writing materials had on the early literacy development of children in early childhood classrooms [38]. After conducting classroom observations and teacher interviews, the authors concluded that more teacher-directed and spontaneous literacy behaviors occurred in the spring semester compared to the fall semester. Furthermore, interviews conducted as part of a qualitative study performed by Jacoby and Lesaux (2019) revealed that teachers in Head Start programs in the northeastern United States believe that English language acquisition occurs naturally within the preschool setting and that social-emotional competencies are foundational for this [39].

However, more studies integrating quantitative and qualitative methodologies are needed to capture the diverse ways in which preschool teachers use language and interact with children. A mixed methods approach can help us to better capture the teachers’ language practices in multilingual preschool classrooms and determine the breadth and depth of teacher language use. The combination of quantitative and qualitative data allows for a more complete understanding of the investigation than either approach by itself [40] and will be used in this study. Centering the teachers’ voices regarding their use of language in the classroom is crucial in understanding the rationale behind their choices and for closely examining the educational settings wherein multiple languages and cultures coexist.

Of the approximately 1,000,000 children enrolled in Early Head Start and Head Start programs, at least 308,750 primarily speak a language other than English at home [41]. Despite the Office of Head Start’s commitment to “support children who speak languages other than English at home in ways that are culturally and linguistically responsive” [6], there remain gaps in the literature concerning the language use of teachers with DLLs in Head Start. Exposure to multiple languages in childhood can assist with language development and potentially foster more effective communication [42,43]. Filling in these gaps is critical in informing Head Start program curriculums and future research directions on the DLL population. Therefore, the current study investigated the following research questions using quantitative and qualitative data and a variety of analyses:What are the characteristics of teachers’ language backgrounds and language proficiency in Head Start classrooms? What are the frequencies of teachers’ English and/or HL use with DLLs across classroom activities?Is teachers’ relative frequency of English vs. HL use in the classroom associated with DLLs’ English and HL expressive vocabulary when controlling for family and child demographic factors (child’s age, gender, generational status, length of preschool school attendance, and family SES) and home language exposure, as well as teachers’ education, years of teaching experience, and English and HL proficiency? We hypothesize that teachers’ higher English use in the classroom would be positively associated with higher English expressive vocabulary scores among participating DLLs. Similarly, teachers’ higher HL use in the classroom would be positively associated with higher HL expressive vocabulary scores among participating DLLs.How do bilingual teachers make meaning of their language choices when interacting with DLLs?

The quantitative data from the teacher surveys were used to address the first question. The quantitative data from the teacher and parent surveys and child language assessments were used to address the second question. Finally, the qualitative data from the teacher interviews were used to address the third question.

## 2. Materials and Methods

### 2.1. Participants

Participants of teacher surveys. The teacher survey sample consisted of 53 teachers from 26 Head Start and 2 state-funded preschool centers serving large proportions of DLLs in urban and suburban areas of Northern California. The teachers were recruited for a larger study on the bilingual and socio-emotional development of preschool-aged DLLs from Chinese-American and Mexican-American families. To recruit teachers and families, we first selected Head Start or state-funded preschool centers serving large proportions of DLLs from Chinese- or Spanish-speaking families. Project coordinators then contacted the directors of the preschool centers to invite them to participate in the study. After creating partnerships, bilingual research assistants visited the centers and recruited interested teachers by distributing flyers and consent forms. Teachers were compensated USD 25 for completing the survey. The teachers’ demographics, language background and proficiency, and years of teaching experience are reported in Table 1.

Participants of teacher interviews. Three teachers were selected from the larger teacher sample and interviewed by the first and second authors. The three teachers were selected based on the following criteria: (1) they worked at our target preschool centers; (2) they worked with DLL children in classrooms; (3) they had varying years of teaching experience—we intentionally selected teachers with diverse lengths of teaching experience; and (4) they had various language profiles. While all teachers were fluent in English, we selected teachers with fluency in different heritage languages (Spanish, Cantonese, and Tagalog). The main characteristics of the teachers that participated in the interviews are shown in Table 2.

Ms. Liu (pseudo-name assigned for confidentiality and anonymity purposes) is an educator who mostly teaches Chinese-American children attending a state-funded preschool in an urban area in Northern California. She is a female and speaks fluent English and Cantonese. Ms. Gonzalez (pseudo-name) is a Head Start teacher who mostly teaches Spanish and Farsi-speaking students and families in another urban area. She is a female and is fluent in English and Spanish. Ms. Valbuena (pseudo-name) worked at an urban Head Start center serving mostly children from Mandarin, Cantonese, and Tagalog-speaking families. She is a female and speaks English and Tagalog.

Participants of child assessments and parent surveys. DLLs and their parents were recruited from the same Head Start and state-funded preschool centers as the teacher participants. Bilingual research assistants visited the sites, distributed project flyers to parents, and collected parents’ contact information. They later contacted the parents to screen the child’s eligibility. The eligibility criteria were as follows: (a) the child is between 36–71 months old; (b) the child attends a Head Start or state-funded preschool at least 3 days per week; (c) the child understands and speaks some English and Cantonese, Mandarin, or Spanish; and (d) both parents self-identify as ethnically Chinese or Mexican. Since the larger project focused on typically developing children, children who were diagnosed with a speech or language disorder or were receiving speech or language services were excluded from the study. The project was described as a research study to understand the linguistic and emotional development of children in Chinese-American and Mexican-American families. The child assessment and parent survey were administered during a 2.5–3.0 h home visit that consisted of one-on-one child assessments, parent interviews and surveys, and parent–child interaction tasks. Upon completion of assessments, children received a small prize and parents received USD 100 compensation for their participation.

The child sample consisted of 190 children (47.9% girls, age range = 36–63 months, M age = 48.3 months, and SD = 7.2) from Spanish-speaking Mexican-American families (MA; N = 65) and Chinese-speaking—mostly Cantonese (N = 72) and some Mandarin (N = 8), too—Chinese-American families (CA; N = 125). On average, the children attended a Head Start or state-funded preschool for 28.2 months (SD = 12.4; range = 2–53 months); the children’s first exposure to English was at 4.9 months (SD = 8.7; range = 0–48 months). The children’s first exposure to a heritage language (Spanish or Chinese) was at 0.55 months, on average (SD = 2.9; range = 0–36 months). Of the children, 12.0% were first-generation (born outside of the U. S.), 82.6% were second-generation (born in the United States and had at least one foreign-born parent), and the remaining 5.4% were third-generation or above (born in the U.S. and both parents were U.S.-born). For each participating child, one parent was asked to fill out the parent survey. Of the participating parents, 96.8% were biological mothers, 1.1% were adoptive mothers, and 2.1% were biological fathers. On average, the parents had lived in the United States for 12.6 years (SD = 7.4; range = 3–40 years). Parents indicated their levels of highest education on an 11-point scale (from 0 = did not attend school to 10 = doctorate). The mean level of parental education was 4.3 (between high school graduate/GED and some college/vocational/technical school). The families’ average per capita income in the past year (calculated as total family income divided by household size) was USD 8994.8 (SD = USD 6676.96; range = USD 625 to USD 32,500).

### 2.2. Procedures and Measures

Teachers and parents provided informed written consent before participating. Parents also provided written permission for their children to participate in the study. All study procedures were approved by the Institutional Review Boards (IRB) of the authors’ institutions.

#### 2.2.1. Teachers Surveys

Teachers completed the surveys online or by mail. The teacher survey was adapted from the BIOS-School (Bilingual Input-Output Survey) [44] and Teacher/Childcare Provider Questionnaire [45]. The BIOS-School questionnaire included detailed questions about their classroom’s language use during a typical school day on an hour-by-hour basis. The Teacher/Childcare Provider Questionnaire included one item inquiring into teachers’ general use of language(s) with DLLs in the classroom/group (e.g., “What languages do you usually talk to the DLLs?”). Moreover, three items inquired into teachers’ use of language(s) in instructional activities, small group activities that include DLLs, organized whole-group activities involving going outside or preparing for lunch, etc. The teachers responded to the items on a 5-point scale (from 1 = All Spanish/Chinese/other HL, 2 = Mostly Spanish/Chinese/other HL, 3 = Same amount of English and Spanish/Chinese/other HL, 4 = Mostly English, and 5 = All English). Additionally, teachers answered brief questions about their ethnicity, language history, and language abilities (with respect to understanding, speaking, reading, and writing) in English and HL when applicable (e.g., Arabic, Cantonese, Farsi, Hmong, Mandarin, Spanish, Tagalog, Taishanese, Urdu, and Vietnamese), as well as their years of teaching experience working with DLLs.

#### 2.2.2. Child Language Assessment and Parent Survey

During the home visit, each child was individually administered the Picture Vocabulary subtest from the Woodcock–Johnson IV Tests of Oral Language [46]. In this 44-item subtest designed to elicit children’s expressive vocabulary, children were asked to name the pictures shown by the assessor with raw scores ranging from 0–44. Validated parallel measures were available in English and Spanish. For Chinese, we adapted the pictures from the Spanish version, as has been performed in previous studies [47,48,49]. The median test reliability for English at age 4 is 0.94 [50] and for Spanish at age 4 is 0.89 [51]. The alpha reliability in Chinese for our sample was 0.91. Raw scores were used for analyses.

During the home visit, one parent was individually administered a parent survey in their preferred languages (Spanish, Traditional or Simplified Chinese, or English). Adapted from prior studies [52], the survey included items regarding the child’s age, child’s gender, child’s age upon first exposure to English, parent’s age, parent’s country of birth and length of stay in the U.S., race/ethnicity of parents and children, parents’ education, family annual income, and household size. In addition, to assess the child’s home language exposure, parents responded to questions about the language spoken most frequently between various dyads in the home (among adults, mother to child, father to child, and other adults to child). Parents rated language use among each dyad using a 5-point scale (1 = only Chinese/Spanish, 2 = Chinese/Spanish and English but more Chinese/Spanish, 3 = Chinese/Spanish and English equally, 4 = Chinese/Spanish and English but more English, and 5 = only English). The alpha reliability for home language exposure across dyads was 0.77. Thus, a composite score of home language exposure was computed by averaging the item scores, with 1 indicating only HL exposure and 5 indicating only English exposure.

#### 2.2.3. Teacher Qualitative Interviews

Three teachers were recruited from the larger sample of 53 teachers who had answered the questionnaire to perform a semi-structured interview designed and conducted by the first and second authors of this study. Three open-ended questions regarding language choices were asked during the interviews (see Appendix A). To minimize the effects of unreliability of memory, the interviewers asked participants to think back to and recall specific moments from their teaching practices. We also asked teachers to provide concrete examples from their lessons and interactions with the students. Teachers were compensated with a USD 100 gift card for their time.

### 2.3. Data Analysis Plan

To address research question 1 (characteristics of teachers’ language use), descriptive statistics (frequencies) were computed from the teacher survey items. To address research question 2 (links of teacher classroom language use to child vocabulary), two random-intercept models were specified to predict children’s English and HL expressive vocabulary from the following predictors: (a) child/family level (level-1) predictors, namely, culture group (CA vs. MA), child’s gender (females vs. males), child’s age, child’s generational status, child’s age upon first exposure to English (or HL), child’s years in Head Start or state-funded preschool, home language environment, and family SES index, and (b) teacher/classroom level (level-2) predictors, namely, teacher’s general classroom language use with DLLs, teachers’ education level, and teachers’ years of experience teaching DLL students. The family SES index was computed by first averaging maternal and paternal education and then averaging the standardized scores of parental education and per capita income. SPSS 27.0 [53] was used to conduct the descriptive analyses and test the random-intercept models (using MIXED command). To address research question 3, the three interviews were transcribed and then analyzed following axial coding (or analytical coding) [54] to identify recurring regularities in the data and segments of the data that are responsive to the study objectives [55]. The emerging themes identified are explained below.

## 3. Results

### 3.1. Research Question 1: Characteristics of Teachers’ Language Background and Language Use

Teachers’ language backgrounds. As reported in Table 1, from the teacher survey sample, 2 (3.8%) teachers were monolingual, whereas 22 (41.5%) teachers were bilingual and 29 (54.7%) were multilingual (spoke 3 or more languages). Both monolingual teachers spoke English. Of the bilingual teacher sample, seven (31.82%) teachers spoke English and variants of Chinese (Cantonese, Mandarin, and Taishanese), six (27.27%) teachers spoke English and Spanish, and nine (40.91%) teachers spoke English and languages including Tagalog, Hmong, Vietnamese, Arabic, and French. Of the multilingual teacher sample, 3 (10.34%) teachers spoke English, Spanish, and “other languages,” and 23 (79.31%) teachers spoke English and at least 2 variants of Chinese. One (3.45%) multilingual teacher spoke English, Cantonese, and Vietnamese. Two (6.9%) multilingual teachers spoke English and other languages (e.g., Punjabi, Hindi, Tamil, and Telugu).

Teachers’ language use throughout a typical school day. We examined teachers’ hour-by-hour language use on a typical preschool day. See Table 3. Two teachers did not report their language use throughout a school day; thus, the sample size was 51 for this analysis. As reported in Table 4, 26 teachers (51%) used both English and other languages 50% of the time, 12 teachers (23.53%) spoke English all the time, 1 teacher (1.96%) used other languages all the time, and 11 teachers (21.57%) used English more than other languages. Only one teacher (1.96%) used other languages more than English.

Teachers’ language use across classroom activities. We further asked the teachers about their language use during specific classroom activities throughout the day. The 22 bilingual and 29 multilingual teachers were asked to describe their language use in several instructional situations. As reported in Table 4, when talking to DLLs, 50.94% of the teachers reported using the same amount of both English and other languages; a total of 1.89% used mostly other languages, 26.42% used mostly English, and 20.75% used purely English. During large group activities, 26.42% of teachers used both languages the same amount, 37.74% used mainly English, and 33.96% used English only. In small group activities, 43.40% of teachers used both languages for the same amount of time, 20.75% used mostly English, and 33.96% used only English. During outdoor activities, 37.74% of teachers used both languages for the same amount of time, 33.96% used mostly English, and 26.42% used only English. A total of 1.89% of teachers used other languages only. In general, the teachers appear to equally use two languages more often when speaking with DLLs individually. However, in group settings, the teachers used a more varied combination of English and their second language.

### 3.2. Research Question 2: Links Regarding Teachers’ General Classroom Language Use to DLLs’ English and HL Expressive Vocabulary

Since this is a community-based sample and children were clustered by classrooms/teachers, there were unequal cluster sizes across classrooms and teachers: each teacher had between 1 and 6 students participating in the study and the average cluster size was 3.6.

The model predicting Child’s English expressive vocabulary. As shown in Table 5, among the level-1 predictors, a child’s age was a significant predictor of an English vocabulary, such that older children scored higher on English expressive vocabulary than younger children (b = 0.37; *p* < 0.001). The home language environment was a significant predictor, such that more English (vs. HL) exposure at home was associated with children’s higher English expressive vocabulary (b = 1.71; *p* = 0.017). In addition, a child’s generational status was a marginally significant predictor, such that later generations (e.g., 2nd and 3rd generations) scored higher on English expressive vocabulary than earlier generations (e.g., 1st generation), b = 2.32, *p* = 0.056. However, none of the level-2 predictors (teacher language, education, and teaching experience) were significantly associated with children’s English vocabulary when controlling for child- or family-level variables.

The model predicting children’s HL expressive vocabulary. Among the level-1 predictors, home language environment was a significant predictor, such that greater English (vs. HL) exposure at home was associated with children’s lower HL expressive vocabulary (b = −2.51; *p* = 0.005). Family SES was a significant predictor such that children in families with a higher SES index scored higher on HL expressive vocabulary than those from families with a lower SES index (b = 1.52; *p* = 0.037). Furthermore, a child’s generational status was a marginally significant predictor such that later generations (e.g., second and third generations) scored lower on HL expressive vocabulary than earlier generations (e.g., 1st generation) (b = −2.66, *p* = 0.065). Among the level-2 predictors, only teachers’ years of teaching experience was a significant predictor, such that the children whose teachers had more years of teaching experience with DLLs scored higher on HL expressive vocabulary than those children whose teachers had fewer years of teaching experience with DLLs (b = 0.18: *p* = 0.041).

### 3.3. Research Question 3: Teachers’ Meaning-Makingof Language Use with DLLs

After performing axial coding, the data from the three interviews revealed several similarities regarding the teachers’ language practices in preschool classrooms. Three main themes were identified: heritage language maintenance, code-switching and translation, and other linguistic and paralinguistic features.

Heritage Language Maintenance. Heritage language maintenance was a shared goal among the three teachers. Ms. Gonzalez, for example, explained that when children have strong skills in their HL, they can use those skills to learn English, even if that means using Spanish or Farsi most of the time in the classroom. Ms. Gonzalez said that she usually gives instructions in English and Spanish, but because her proficiency in Farsi is low, she usually asks children who speak both Farsi and English to repeat the instructions in their HL for those who only speak Farsi. Similarly, Ms. Valbuena shared that, in her preschool, teachers always encourage HL use and do not usually intervene in children’s language choices. Moreover, in her interview, Ms. Liu said: “We promote home language, so we speak in their home language, but when we are introducing lessons, we do speak both in Chinese and English. And we don’t favor English in comparison to the home language. [...] We wouldn’t say ‘You have to say it this way’, we would give them the choice. Like, for example, an apple, then they would say ‘Ping Guo’ if they want. We let them have the choice [...] whether they understand it in Cantonese or if it’s in English”.

The interview data also indicated that teachers who do not share the HL with some children still transcend their regular duties to create a supportive environment. In addition to adopting the strategies described above, the teachers shared that they usually slow down or even learn some words in their students’ HL. Ms. Gonzalez, for instance, explained: “I learned some Farsi, which is really hard. But I try, so the children feel that they belong, so it’s for both their language and social-emotional [development]. [...] When they start picking it up, I do it like, super slowly and one word at a time.” Ms. Liu also stated: “We have some children who speak Taishanese, so we try to speak slow, because none of our teachers speak Taishanese, we are just able to speak in Cantonese and Mandarin back to the children”.

Code-switching and translation. Frequent code-switching (both at the word level and sentence level) and translation practices were also reported by the teachers interviewed. Ms. Valbuena, who works with primarily Cantonese and Mandarin-speaking students and speaks English and Tagalog herself, said that she often asks assistant teachers to translate for her if a child does not understand English. She explained that teachers always start by speaking English, and if a child does not comprehend this, then the teachers switch to the student’s HL if they have the language ability. If a teacher does not speak the student’s HL, they ask for translational help from assistant teachers.

The interview data revealed that translation practices were particularly frequent during book-sharing activities in the classroom. For example, Ms. Liu said: “It’s during the lesson, you would translate. We do read in both Cantonese, Chinese, and English”. Ms. Gonzalez described to us that she only speaks Spanish during the lesson when a child is Spanish dominant, but she always reads books in both Spanish and English. Similarly, during her interview, Ms. Gonzalez said: “Yes, sometimes we read books in both languages. Particularly if I have like a big number of children that are Spanish speakers [...] usually, the books I read in Spanish are short, so I will say it in English and in Spanish, but the book would be in Spanish, then I will repeat it in English, for those who do not speak English”.

Other linguistic and paralinguistic features. During the interviews, all three teachers provided rich and varied examples of other forms of linguistic and paralinguistic support offered to DLLs in their classrooms. To support children’s language- and literacy-learning capacities, Ms. Liu, Ms. Gonzalez, and Ms. Valbuena reported using a variety of visuals, modeling, and gesturing. They also explained that they frequently ask students comprehension questions, in addition to defining terms and objects, and using repetition. Ms. Liu told us the following: “So in the class, we do have visuals. Another technique we use is using picture interaction among other children. We use gestures, and then just repeat the word, both in Cantonese, Mandarin, or in English, and a lot of body language”. Similarly, Ms. Gonzalez explained: “I usually use a lot of signs and a lot of visuals [...] Because, and you know, learning a few words in their own language and trying to communicate that way but mostly it’s with pictures and then modeling a lot”.

## 4. Discussion

The purpose of this paper was to examine Head Start teachers’ backgrounds and language use and how these facets were associated with DLLs’ language skills, and to further understand how bilingual teachers make meaning of their language choices when interacting with DLLs. Regarding the teachers’ backgrounds, years of experience, and language abilities, the teacher survey results showed that most of the teachers sampled in our study were multilingual, with over 90% speaking more than one language, which were typically English and a Chinese dialect. Over 80% of the teachers identified as Asian. More than half of the preschool teachers had a bachelor’s degree or higher and were experienced early childhood educators with an average of 15.63 years of teaching, including an average of nearly 10 years of teaching DLLs. This has positive implications, especially for DLLs who come mostly from low-SES families and attend schools with a highly diverse ethnic composition [22], given that a more diverse and experienced teacher body can aid cultural competence, language development, and identity development for children of immigrant parents [56]. As explained above, having preschool teachers who can speak and understand the children’s HL is important for DLLs’ language development, especially for DLLs who have higher fluency in their HL than in English. Additionally, DLLs with higher fluency in HL tend to interact more with teachers who are culturally competent. This is because these teachers share their HL and ethnic background.

The results of the random-intercept models did not support the idea that teachers’ language use when addressing DLLs in the classroom predicts children’s expressive vocabulary. This finding contradicts past research. Ramirez and colleagues (2020) found that teacher talk in DLLs’ (3–8 years old) home language contributed to their language development in both English and their heritage language [57]. Additionally, Collins (2014) found that Latino children in kindergarten and second grade who had received instruction in English and Spanish showed higher gains in both languages than Latino children whose teachers spoke only or mostly English [36]. Previous research on older children has suggested that encouraging bilingual teachers to use their HLs to supplement English instruction may be effective in developing children’s language and literacy abilities [58,59]. This contradiction could be due to our sample’s age group. The children who participated in our study were younger (3–5 years old). The benefit of teachers’ English and HL use could be long-term and, therefore, be undetectable by our cross-sectional study. Future research should aim to investigate how teachers’ language use in preschool classrooms contributes to DLLs’ success in later school years.

In addition, fostering a supportive environment for DLLs in the classroom may encourage them to express themselves more openly and alleviate external pressure to speak English. To create a supportive environment, some teachers in our study learned words in the DLLs’ home languages. This aligns with some of the teaching practices reported by Baker and Páez (2018), who also shared other strategies including asking families to contribute resources in their home language [37], labeling classroom areas and materials, and incorporating home languages into classroom routines. Therefore, even when teachers do not know some of their students’ home languages, there are other resources that they can use to expose DLLs to their HL in meaningful forms. Languages are interconnected in bilingual brains and one language is dependent on the other to make sense of the world [60]. Indeed, one of the interviewed teachers explicitly acknowledged that learning an HL can help children acquire English. This is also supported by recent research in a Head Start center in the northeastern United States, where teachers reported that using Spanish in the classroom is a strategy that supports socio-emotional development and facilitates DLLs’ English language acquisition [39].

Prior research suggests that children’s exposure to bilingual instructional models can improve their academic outcomes, English acquisition, and HL development [24,61,62,63,64]. Despite the sampled DLLs’ low oral English vocabulary knowledge, they can have conceptual knowledge similar to their monolingual peers [65,66]. Additionally, the English vocabulary levels of DLL children in programs with bilingual instruction do not seem to differ from their DLL peers in English immersion classrooms [67,68]. However, these studies have been primarily focused on the Spanish-speaking DLL population or the academic achievement of older bilingual children. Our unique multilingual sample of teachers and our emphasis on preschool-aged children reveal further relationships between language and academic achievement, warranting more in-depth investigations. Prior research has also shown that when working with both DLL and non-DLL students, lead and assistant teachers predominantly use English as their language of interaction [69]. Yet, our study showed that teachers used both HL and English in classrooms with DLL children. Thus, more research is needed to better understand the sources of contextual and individual variabilities in teachers’ language use in DLL preschool classrooms and their links to children’s developmental outcomes.

Our findings also revealed that, as expected, home English (vs. HL) language exposure was positively associated with a child’s English vocabulary, while being negatively associated with HL vocabulary. Additionally, a child’s generational status was also positively associated with a child’s English vocabulary, while being negatively associated with HL vocabulary. Furthermore, the children’s age was also positively associated with children’s English vocabulary, while family SES was associated with children’s HL vocabulary. These findings support the idea that the environment—both home and school—in which children develop is of crucial importance for their language development. Elements of the home and school environment as microsystems [22,70] can hinder or enhance DLLs’ English development and HL maintenance. Additionally, our interview data revealed that code-switching and translation practices along with gesturing, the use of visuals, and repetition are some successful strategies that preschool teachers use in multilingual classrooms.

The current mixed methods analysis was used for the purposes of complementarity and the development of findings from quantitative and qualitative data [40]. We used the initial teacher survey to gain knowledge about teachers’ language backgrounds and language use in multilingual classrooms. The survey results informed our open-ended questions for the qualitative interviews. For instance, since the survey questions informed us that teachers frequently alternate between languages in the classroom, in our interviews, we asked about their rationales behind language choices such as switching between languages. The interview data allowed us to gain a deeper understanding of specific situations in which preschool teachers use a range of strategies to communicate with DLLs. Thus, the mixed methods design strengthened the validation and interpretations of our findings.

Our findings provide a preliminary glimpse into the varied language profiles of teachers in Head Start preschool programs and have implications for preschool classrooms and teachers’ professional development. These results suggest that teachers can benefit from training sessions wherein more experienced teachers share the challenges faced when working with DLLs and successful practices and approaches. Moreover, sharing effective classroom strategies will also be helpful for new and experienced preschool teachers in culturally and linguistically diverse classrooms.

### Limitations

The study has a few limitations. First, we relied on self-reported surveys and interviews to assess teachers’ oral language use in classrooms. High-quality, rich, and varied linguistic experiences may also include visual stimuli such as books, signs, and logos in the classrooms, which our study was unable to examine. Furthermore, observation methods are crucial to understanding actual classroom language practices. Future studies should consider incorporating multiple methods and measures to assess children’s language experience and teachers’ language use in multilingual classrooms. Second, the study was conducted with a convenience sample of teachers and DLLs in Head Start programs in Northern California, which is unique with respect to the sociocultural and socioeconomic diversity in its population. Thus, the results might not be representative of early childhood educators in other programs or geographic areas.

## 5. Conclusions

The success of the DLL population requires specialized training and attention from teachers and school administrators. Teachers’ possession of greater amounts of training, years of experience working with DLLs, and cultural competency have all been shown to positively affect children’s language outcomes [29]. However, as our data revealed, the amount of training and experience that DLL teachers receive and bring to the classroom is far from uniform. Procedures and programs to help standardize teachers’ training and familiarize instructors with the DLL demographic should be researched and developed.

The bilingual characteristics of teachers have implications that extend outside the classroom. The maintenance of a multicultural and multilingual school setting is attractive to parents. In previous studies, parents have been shown to tend to enroll their children in centers where providers speak their children’s HL, preferring a school that is culturally close to their own [59,71]. Promoting a bilingual environment at Head Start centers also increases families’ involvement in school, leading to stronger relationships between students and teachers, and promoting improved academic competency and attitudes in children [72]. Bilingual educators would also be able to understand and fulfill parental perspectives on bilingual education. Some parents may express concerns about bilingual education slowing down their child’s English language acquisition or causing possible confusion in a multilingual classroom. Bilingual teachers and educators are key in engaging parents with these conversations and addressing their concerns while respecting parents’ perspectives. The influence of language on parental behavior and student–teacher relationships poses directions that could be investigated further.

The DLL population continues to grow in the United States and, as we progress towards an even more diverse demographic, researchers must expand their studies on this population to better understand DLLs’ learning mechanisms and how bilingualism contributes to their success in preschool and later school years.

## Figures and Tables

**Table 1 children-09-01871-t001:** Teachers’ demographics, language proficiency, and years of experience (*N* = 53).

Geographic Areas of Preschool Centers	%				
Urban city	58.49%				
Suburban area	20.75%				
Urban area	20.75%				
Teacher gender					
Male	5.66%				
Female	94.34%				
Teacher origin					
Not of Hispanic, Latino, or Spanish Origin	86.79%				
Mexican, Mexican-American, Chicano	7.55%				
Of another Hispanic, Latino, or Spanish Origin	5.66%				
Teacher race					
White	13.21%				
Black or African-American	3.77%				
Asian	79.25%				
Other	3.77%				
Teacher language profile					
Monolingual	9.43%				
Bilingual	39.62%				
Multilingual	50.94%				
Teacher education					
High School/GED	1.92%				
Some College	15.38%				
Associate degree	23.08%				
Bachelor’s Degree	44.23%				
Some Graduate School	3.85%				
Master’s or more advanced degree	11.54%				
Teachers’ English language proficiency		M	SD	Min.	Max.
Understanding spoken language		3.47	0.54	2	4
Speaking		3.40	0.63	2	4
Reading		3.45	0.54	2	4
Writing		3.34	0.65	2	4
Teachers’ Heritage language proficiency					
Understanding spoken language		3.82	0.45	2	4
Speaking		3.74	0.59	2	4
Reading		3.62	0.91	1	4
Writing		3.51	0.97	1	4
Years of teaching experience with children					
Any age		15.63	8.82	2	37
At the current program/center		8.88	7.76	0	28
Experience with infants/toddlers (<age 3)		2.60	3.72	0	16
Experience with preschoolers (age 3–5)		11.38	9.48	0	37
Experience with dual language learners		9.92	7.74	0	28

**Table 2 children-09-01871-t002:** Characteristics of the teachers participating in the interviews (*N* = 3).

	Ms. Liu	Ms. Gonzalez	Ms. Valbuena
Preschool location	urban	urban	urban
Population served	Chinese-American	Mexican-American Afghan-American	Chinese-American Philippine-American
Years of experience with children (any age/grade)	6	12	25
Years of experience with preschoolers (3–5-year-olds)	4	10	15
Years of experience with DLLs	6	10	2
Language proficiency	English and Cantonese	English, Spanish, and some Farsi	English and Tagalog
Teacher’s ethnicity	Chinese	Mexican	Filipino

**Table 3 children-09-01871-t003:** Frequency of teachers’ language use in the classroom (*N* = 53).

Language Use Towards DLLs	
All English	20.75%
Mostly English	26.42%
Same Amount of English and Spanish/Chinese/other Language	50.94%
Mostly Spanish/Chinese/Other Language	1.89%
All Spanish/Chinese/Other Language	0.00%
Whole-Group Instructional Activity	
All English	33.96%
Mostly English	37.74%
Same Amount of English and Spanish/Chinese/other Language	26.42%
Mostly Spanish/Chinese/Other Language	0.00%
All Spanish/Chinese/Other Language	1.89%
Small Group Activity including DLLs	
All English	33.96%
Mostly English	20.75%
Same Amount of English and Spanish/Chinese/other Language	43.40%
Mostly Spanish/Chinese/Other Language	0.00%
All Spanish/Chinese/Other Language	1.89%
Whole Group Outside/Lunch	
All English	26.42%
Mostly English	33.96%
Same Amount of English and Spanish/Chinese/other Language	37.74%
Mostly Spanish/Chinese/Other Language	0.00%
All Spanish/Chinese/Other Language	1.89%
Language Use Throughout School Day	
All English	23.53%
Mostly English	21.57%
Same Amount of English and Spanish/Chinese/other Language	51.00%
Mostly Spanish/Chinese/Other Language	1.96%
All Spanish/Chinese/Other Language	1.96%

**Table 4 children-09-01871-t004:** Descriptive statistics of child and family variables for the full sample and by cultural groups (*N* = 190).

Variables	Full Sample (*N* = 190)				Mexican-American Group (*N* = 65)		Chinese-American Group (*N* = 125)	
	M	SD	Min	Max	M	SD	M	SD
Child’s age (months)	48.34	7.25	36	63	49.15	6.57	47.91	7.57
Child’s years in Head Start/state-funded preschool	28.18 *	12.35	2	53	32.19	12.31	26.20	11.95
Child’s age upon first English exposure (months)	4.92 *	8.69	0	48	7.58	11.43	3.56	6.53
Child’s age upon first heritage language exposure (months)	0.55	2.92	0	36	USD 9769	USD 6723	USD 7403	USD 4321
Average parent education level ^1^	4.30 *	0.18	0.5	9.5	3.60	2.11	4.64	2.60
Per capita income	USD 8994 *	USD 6131	USD 625	USD 32,500	USD 7403	USD 4321	USD 9769	USD 6723
Home language exposure ^2^	1.54	0.78	1	4.75	1.68	0.93	1.48	0.70
Child’s English expressive vocabulary	11.86	5.88	0	28	11.05	5.74	12.30	5.93
Child’s Spanish/Chinese expressive vocabulary	10.68	6.75	0	23	10.28	7.21	10.88	6.53

^1^ Parental education level was coded on an 11-point scale (from 0 = did not attend school to 10 = doctorate degree); ^2^ home language exposure was coded on a 5-point scale (from 1 = only Spanish/Chinese to 5 = only English); variables with star(s) indicate significant cultural group differences in means based on independent-sample *t*-tests.

**Table 5 children-09-01871-t005:** Random-intercept models predicting child expressive vocabulary from teacher- and child/family-level variables.

	Model Predicting Child’s English Expressive Vocabulary		Model Predicting Child’s HL Expressive Vocabulary	
Fixed Effects	Coefficient (SE)	*p*	Coefficient (SE)	*p*
Intercept	−13.47 (5.27)	0.012	5.96 (6.68)	0.375
Level-2 predictors				
Teacher’s general classroom language use ^1^	0.83 (0.81)	0.309	0.976 (1.04)	0.355
Teacher’s highest education level	−0.25 (0.41)	0.542	0.306 (0.55)	0.583
Teacher’s years of experience with DLLs	0.03 (0.06)	0.600	0.178 (0.08)	0.041
Level-1 predictors				
Child’s age	0.368 (0.07)	0.000	0.148 (0.09)	0.108
Child’s gender (males vs. females)	−0.511 (0.99)	0.608	−1.768 (1.17)	0.133
Child’s generational status	2.320 (1.20)	0.056	−2.66 (1.43)	0.065
Culture group (CA vs. MA)	0.766 (1.39)	0.583	−0.06 (1.85)	0.974
Child’s years of preschool attendance	−0.062 (0.05)	0.210	0.047 (0.06)	0.402
Child’s age upon first English/HL exposure	0.020 (0.08)	0.788	0.037 (0.53)	0.945
Family SES	0.764 (0.59)	0.196	1.523 (0.72)	0.037
Home language exposure ^2^	1.713 (0.70)	0.017	−2.507 (0.88)	0.005

^1^ Teacher’s general classroom language use was coded on a 5-point scale from 1 = All Spanish/Chinese to 5 = All English; ^2^ Home language exposure was coded on a 5-point scale from 1 = only Spanish/Chinese to 5 = only English.

## Data Availability

Data are not publicly available because we did not have consent in favor of public access from the subjects involved.

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
