# Peer review of "Teachers’ Language Use in Multilingual Head Start Classrooms: Implications for Dual Language Learners"

_children, 2022, doi:10.3390/children9121871_

Round 1
Reviewer 1 Report (Previous Reviewer 2)
Good revisions.
Author Response
Dear Reviewer,
Thank you very much for your time and helpful comments that have allowed us to strengthen our manuscript.
Best regards,
Maria B. Buttiler
Reviewer 2 Report (Previous Reviewer 1)
Thank you for inviting me to review the paper “Teachers' Language Use in Multilingual Head Start Classrooms: Implications for Dual Language Learners”. Study is interesting and well written.
Some recommendations and questions are as follows:
1. Chinese – do you mean Mandarin Chinese
2. Home language – is this synonymous to first language
3. Just curious - would the use of bilingual approach and code-switching results in mixed language used, eg. Singapore English “Singlish” a mixed usage of Chinese and English or Tag-lish a mixture of English and Tagalog
4. Note the use of “&” within the text, some should be “and”, eg. Jacoby & Lesaux (2019) – should be Jacoby and Lesaux (2019)
5. Since this is a mixed-method approach – how the quantitative and qualitative data intersect / complement each other should be clear
In sum the study is interesting and difficult to accomplished, the authors obviously did a great job in collecting the data. However, as noted, how the quantitative and qualitative data complement and enriched the later data / results interpretation is also important.
Author Response
Dear Reviewer,
We would like to thank you for the constructive comments and time dedicated to our manuscript. We have made every attempt to address these comments in the revised version of the paper and we believe these revisions have resulted in a significantly improved manuscript.
In the attached manuscript, the comments were made using Track Changes. The following paragraphs summarize our responses to your comments.
- Chinese – do you mean Mandarin Chinese
The DLLs in our sample speak mostly Cantonese (N = 72), and some of them speak Mandarin (N = 8). This has been specified in the Participants of child assessments and parent surveys section on page 6.
- Home language – is this synonymous to first language
Given the diverse linguistic practices of the families and teachers, DLLs may be exposed to either low or significant levels of English as well as other languages. On page 2 of our manuscript, we wrote that DLLs in the U.S. start learning English as a second language while they are still acquiring their first language, also known as home or heritage language.
- Just curious - would the use of bilingual approach and code-switching result in mixed language used, eg. Singapore English “Singlish” a mixed usage of Chinese and English or Tag-lish a mixture of English and Tagalog.
This is an interesting question. We added the following to our manuscript:
Additionally, our interview data revealed that code-switching and translation practices along with gesturing, use of visuals, and repetition are some successful strategies that preschool teachers use in multilingual classrooms.
Unfortunately, we did not examine children’s code-switching in this manuscript, but we hope to in the future.
If teachers’ code-switching practices were leading to children’s code-switching, this should be perceived as an asset. In accordance with previous research, DLLs have complex knowledge of how to fit their two languages into one utterance, showing language-specific syntax early on.
Moreover, code-switching can serve as a compensatory strategy when DLLs do not know the target word in the dominant language.
- Note the use of “&” within the text, some should be “and”, eg. Jacoby & Lesaux (2019) – should be Jacoby and Lesaux (2019)
Two typos have been addressed:
findings from Huang and Kan have implications… (page 2)
part of a qualitative study conducted by Jacoby and Lesaux (2019) revealed that… (page 3)
- Since this is a mixed-method approach – how the quantitative and qualitative data intersect / complement each other should be clear
In our manuscript, we had written that “A mixed methods approach can help us to better capture the teachers’ language practices in multilingual preschool classrooms and understand the breadth and depth of teacher language use. The combination of quantitative and qualitative data allows for a more complete understanding of the investigation than either approach by itself [40], and will be used in this study.” (page 4)
We have added the following to the Discussion section: “The current mixed analysis was used for the purposes of complementarity and development of findings from quantitative and qualitative data [40]. We used the initial teacher survey to gain knowledge about teachers’ language backgrounds and language use in the multilingual classroom. Then, we used their responses to inform our open-ended questions for the qualitative interviews. For instance, since the survey questions informed us that teachers frequently alternate languages in the classroom, in the interviews, we asked about their rationale behind language choices such as switching between languages. The interview data allowed us to gain a deeper understanding of specific situations in which preschool teachers use a range of strategies to communicate with DLLs. Thus, the mixed methods design strengthened the validation and significance of our findings.” (page 14).
Thank you again for your time and helpful comments.
Best regards,
Maria B. Buttiler
This manuscript is a resubmission of an earlier submission. The following is a list of the peer review reports and author responses from that submission.
Round 1
Reviewer 1 Report
First of all, thank you for inviting me to review the paper “Teachers' Language Use in Multilingual Head Start Classrooms: A Mixed-Methods Study”. Paper discussed the potential contributions in multi-language / dual-language (medium of instruction) in preschool classrooms.
Some comments and recommendations are provided:
1. Please move the research objectives or research questions (preferably at the end of the introduction and not in the method section)
2. Provide a background theoretical framework of the study
3. Both qualitative and quantitative data were used – should clarify why mixed-method is needed. How does the intersection of the qualitative and quantitative data enrich the findings?
Did the findings provide grounds for the link between language use and academic achievement? Essentially, the present study is descriptive in nature, rather than theoretical or based on deeper assumptions. I am not sure whether the contributions are significant. Additionally, the author/s should note that there are some disadvantages associated with mixing the English language with the heritage language. (such as in the like of “singlish”…).
The use of heritage language for interaction with students would certainly promote better student-teacher engagement. There is, however, a need for further empirical findings to justify the conclusions. Three interview participants seems to be an insufficient number.
Reviewer 2 Report
The topic of exploring teachers' language use in the classroom is timely. However, this submission requires substantial modification for publication. First, the title is misleading. It is a qualitative study with some descriptive statistics - it is not a mixed methods paper in the strict sense of the meaning. Second, the use of all self-report data and interviews is not clearly articulated until you reach the limitations section. For example, many studies show the fallibility of memory when asking people to remember to record items while conducting another task. Third, in the limitations, a discussion is presented on what a teacher would do if the teacher did not speak the heritage language. This is a more significant issue regarding the lack of heritage language speakers in any given school. As one example, the teacher does not know if the child is repeating the correct directions. This line of thinking also brings up another issue of not attending to the different cultures in the school. Some cultures would shun the idea of a child becoming a teacher when the child's role is to learn. The authors should address those issues if they intend to promote students as teacher models for DLLs. Finally, while this study holds promise for future iterations, the authors need to ensure they are not conflating the research they cite with their intentions and fully understand they are looking at a particular set of teachers, which is not common across US public school settings.
Reviewer 3 Report
Thanks for inviting me to review this manuscript, which reports a mixed-methods study of teachers' language profiles in Head Start preschools. Generally, this study is very descriptive and preliminary, contributing nothing important to the theoretical development or practical improvement. In particular, I have some major concerns as follows.
First, the topic. This study focuses on teachers' language use in Head Start classrooms, which has been extensively explored by other scholars. For example, Dickinson, D. K., Hofer, K. G., Barnes, E. M., & Grifenhagen, J. F. (2014). Examining teachers’ language in Head Start classrooms from a Systemic Linguistics Approach. Early childhood research Quarterly, 29(3), 231-244. Baker, M., & Páez, M. (2018). The Language of the Classroom: Dual Language Learners in Head Start, Public Pre-K, and Private Preschool Programs. Migration Policy Institute. Therefore, I am not convinced that this study will have a significant contribution to the literature.
Second, the justification. The introduction and the literature review failed to justify this study. I am not convinced that this study is needed.
Third, the research questions. According to the APA Publication Manual, mixed-methods study papers should have research questions, which are missing in the current version. The quantitative part could also have a hypothesis.
Fourth, the sample size. As a quantitative study, the survey part should have a large size of sample, and the sampling approach should also be well planned. This study only has 53 teachers, which might not be adequate or representative. Therefore, the findings might not be reliable.
Last but not least, the discussion is very superficial and descriptive. Without any significant findings or theoretical framework, this discussion is not in-depth or insightful. In addition, Bronfenbrenner's theory might not be specific or fit. You need to cite some DLL theories.
In conclusion, this study is very premature and preliminary.
Reviewer 4 Report
The present study fills an important gap. Very little is known about the use of languages other than English for early childhood teachers working in settings where English is the official/dominant language. The study includes descriptive details of 53 Head Start teachers, then probes further using interviews with three teachers. Although the design is modest and has limitations (clearly acknowledged by the authors), the study does provide an important new perspective. The authors have been generous in writing and have sign-posted areas that can be extended in research. It is a shame we know so little about teacher language use with preschoolers given the acknowledged role of teachers in promoting language development.
The reporting is clear and overall the article is well written. Citations and references need to be revised to MDPI style - this is my only suggested revision. I also appreciate that the authors have made great efforts to help readers understand the research in the context of the limitations of the study. The reporting is very clear.
I believe the present study could play an important role in accelerating understanding of teacher language in multilingual ECEC settings.